# Evaluation of a CNN-Based Modular Precision Sprayer in Broadcast-Seeded Field

**DOI:** 10.3390/s22249723

**Published:** 2022-12-12

**Authors:** Paolo Rommel Sanchez, Hong Zhang

**Affiliations:** 1Mechanical Engineering Department, Henry M. Rowan College of Engineering, Rowan University, Glassboro, NJ 08028, USA; 2Agribiosystems Machinery & Power Engineering Division, Institute of Agricultural and Biosystems Engineering, College of Engineering and Agro-Industrial Technology, University of the Philippines Los Baños, Laguna 4031, Philippines

**Keywords:** precision agriculture, agricultural sprayer, convolutional neural networks, machine vision, modular robot, agricultural robot, weeding, broadcast-seeded, soybean

## Abstract

In recent years, machine vision systems (MVS) with convolutional neural networks (CNN) for precision spraying have been increasingly investigated due to their robust performance in plant detection. However, the high computational requirement of CNNs makes them slow to be adopted in field operations, especially in unstructured working environments such as broadcast-seeded fields. In this study, we developed a modular precision sprayer by distributing the high computational load of CNN among parallel low-cost and low-power vision computing devices. The sprayer utilized a custom precision spraying algorithm based on SSD-MobileNetV1 running on a Jetson Nano 4 GB. The model achieved 76% ${\mathrm{mAP}}^{0.5}$ at 19 fps for weed and soybean detection in a broadcast-seeded field. Further, the sprayer targeted all weed samples and exhibited up to 48.89% spray volume reduction with a typical walking speed up to 3.0 km/h, which was three times faster than similar systems with known targeting performance. With these results, the study demonstrated that CNN-based precision spraying in a complex broadcast-seeded field can achieve increased velocity at high accuracy without needing powerful and expensive computational hardware using modular designs.

## 1. Introduction

Soybean (*Glycine max* L.) is a widely cultivated crop in the world due to its rich oil and protein content [1]. These crops are commonly planted in rows [2,3,4], but broadcast seeding is also being practiced due to its low labor requirements at minimal yield loss. For example, in the study of Whaley and Uddin [5], they demonstrated that broadcast seeding resulted in only 8% yield loss, but at less labor compared to line-sowing. Similarly, in a recent study by Vandeplas et al. [6], broadcast-seeded soybean with single-pass weeding required only 11% of the planting time of hill-drop-seeded soybean at the expense of 15% yield reduction.

In any field cropping system, weed control plays an essential role in preventing yield loss by minimizing unnecessary resource competition of crops with unwanted plants. Thus, during the last 70 years, chemical-based weed control has become a common component of modern crop production [7]. Nonetheless, despite the increasing environmental and health concern surrounding herbicide usage in recent years [8], the lower labor demand of chemical over mechanical weed control remains the reason for its wide adoption [9]. Further, the absence of distinct row-spacing makes inter-row cultivating and mechanical hoeing inapplicable due to the risk of damaging the crops in broadcast-seeded conditions [10]. Thus, modern soybean farming often plants herbicide-resistant varieties and controls weeds using chemical herbicides [11,12].

Still, uniform herbicide application is economically inefficient and harmful to the environment [13]. Precision agriculture (PA) seeks to minimize chemical usage by using modern sensors, control systems, actuators, and software and delivering site- and time-specific quantities of inputs. Modern PA equipment often utilizes spectrometric, optoelectronic, and imaging sensors for real-time weed detection [14]. For example, to minimize herbicide usage in soybean, experiments using a machine vision system (MVS) and discrete wavelet transform to analyze soil, weed infestation, and crop zones reduced spray volumes by 48% [15]. Their system was 100% accurate in detecting bare soil regions but only 75% and 47.8% accurate in weed and crop zone detection, respectively. On the other hand, using spectrometric sensors and variable valve control reduced the volume of post-emergence spray by approximately 51% [16]. However, the study also reported that the spectrometric sensor could not differentiate the phenological stages of weeds and soybeans, causing high variability in spray reduction.

The complexity of farming environments limits the commercialization and wide adoption of weed sensing technologies. The high variation in spectral characteristics of weeds and crops at different growth stages and weather conditions makes differentiating between plant species using spectrometric and optoelectronic sensors difficult [17]. Additionally, spectral-based sensing with current approaches cannot reliably identify weeds at sufficiently low weed densities due to inadequate differences between the field spectral characteristics of weed and crop [18]. On the other hand, image-based sensing delivers promising results. However, increasing the robustness of algorithms using complex approaches, such as machine-learning-based segmentation and classification, for image-based sensing generally increases computational cost [14,15]. For example, a precision sprayer for targeting weeds between carrot rows by color-based thresholding of normalized difference vegetation index images could operate up to 4.17 m/s, but without differentiating between weeds and crops [19]. A more recent precision sprayer utilized a support vector machine to classify shape, texture, and color feature vectors from a 4-megapixel RGB-IR camera for in-row weed targeting in carrots but limited the operation to 0.8 m/s [10].

Recently, MVS that utilized Convolutional Neural Networks (CNN) for plant detection have been gaining popularity due to their high detection reliability in dense agricultural environments [20]. For example, past studies showed that CNN-based weed detection in soybean can achieve 65% to 99.5% precision [21,22,23]. Nonetheless, despite the high potential of CNN for robust weed detection, the volume of research that demonstrated the use of CNN for actual precision spraying in soybean was very small and had low targeting accuracy. In the study of Sabóia et al. [23], their precision sprayer for specific targeting of cord grass in row-seeded soybeans using Mask R-CNN and YOLOv3 was only 78% and 20% accurate, respectively. Similarly, precision sprayers with CNN-based MVS operate at lower travel velocities than spectral- or optical-based sensors due to the high computational cost. For example, the CNN-based precision sprayer for soybean operated only at 0.5 m/s [23], compared to 5.6 m/s using spectrometric sensors [16] and 1.17 m/s using MVS with traditional image processing [15]. Developed systems often rely on powerful hardware, such as NVIDIA GTX 1050 [23,24], 1070 Ti [25], and 1080 [26], to achieve sufficient inference speed. In a study that compared fast CNN models in the detection of weed patches in soybeans, the inference times for Faster R-CNN and SSD on a desktop with NVIDIA RTX 2080 Ti were 0.23 and 0.21 s, respectively [21]. Offsetting the computation to a powerful remote server with NVIDIA Tesla M10 connected through a 5G network was also explored to increase the speed of CNN-based precision spraying, but the system achieved a similar 0.25 s overall processing time [27].

The high computational requirement of CNN-based detection using image sensors, despite having high detection performance and robustness compared to spectrometric, optical, and distance sensors, hinders its wide adoption for real-time precision spraying [17]. If the precision sprayer is traveling too fast relative to the inference speed of the CNN, gaps will occur between consecutive frames causing certain plants to be missed [28]. To solve the high computational cost of CNN, various CNN architecture designs and optimizations were proposed and showed promising results. Some of these techniques include using single-stage architectures such as Single Shot MultiBox Detector (SSD) [29] and You Only Look Once (YOLO) [30], implementing depth-wise separable convolutions [31,32], reducing feature reuse within convolution [33], and optimizing trained CNN model using TensorRT [34]. In this research, we implemented modular software and hardware architecture to localize the computational load of TensorRT-optimized SSD-MobileNetV1 among multiple low-power and low-cost hardware for precision spraying. This approach, in effect, would increase the operating travel velocity of CNN-based precision sprayers without the need for powerful desktop- or server-grade systems by having dedicated computational hardware for each capture device. Therefore, this paper evaluates the targeting performance and spray volume reduction of the developed modular agrochemical precision sprayer (MAPS) with CNN-based MVS in broadcast-seeded soybean. The suggested design approach and field testing methods in this study aim to increase the feasibility of precision sprayers with CNN-based MVS and contribute towards sustainable farming environments.

## 2. Materials and Methods

### 2.1. Precision Sprayer

Figure 1 shows the precision sprayer developed at the Mechanical Engineering Department of Rowan University, Glassboro, NJ, USA. A summary of the technical specification of the MAPS is then shown in Table 1. The repeating components of the MAPS were grouped into an assembly called a scalable unit (SU), composed of vision and sprayer modules. The MAPS used in the test had three SUs mounted on a push-type frame and were separated by 0.5 m. The overall cost of all sprayer components shown in Figure 1 was approximately USD 2100.

The general workflow of each SU of the MAPS is illustrated in Figure 2. Each SU was connected to the same local network through Ethernet. Component communication was implemented using a robot operating system (ROS). A graphical user interface (GUI), hosted in the central module (Raspberry Pi 4B 4 GB), was then used by the sprayer operator to manage and monitor each SU.

As shown in the vision module of Figure 2, precision spraying starts by streaming a top view of the soybean plot from an RGB camera (Logitech StreamCam). Then, CNN-based inferencing using a TensorRT-optimized SSD-MobileNetV1 running on a CUDA-capable edge device (NVIDIA Jetson Nano 4 GB) detects and generates bounding boxes of soybeans and weeds in the image, as illustrated in Figure 3. Using a custom algorithm in Python, the coordinates of soybeans and weeds were stored in separate lists for each object class. All bounding boxes in the frame were then tracked accordingly, and their Euclidean travel distances between two consecutive frames were measured. The relative velocity of the sprayer was then estimated based on the elapsed time and the distance traveled by tracked plants between two consecutive processed frames. Finally, knowing the instantaneous distance of the weeds from the sprayer nozzle and travel velocity, the time to trigger the sprayer was calculated and stored in a spray schedule list. Appending spray schedules was performed whenever a weed was detected in a frame. Thus, the MAPS also cleared all elapsed spray schedules within the effective spray region of the previous spraying after a sprayed feedback signal was received. This step filters the spray schedules in the list to prevent multiple spraying on already sprayed weeds.

A parallel process then regularly checks the list for elapsed spray schedules. The process then sends a trigger signal to a USB-connected microcontroller (Arduino Nano) that controls a relay switch (Arceli KY-019) if a time value in the spray schedule has been reached. This event causes the solenoid valve (US Solid USS2-00006) to open for 0.2 s and delivers liquid to a fan-type nozzle (Solo 4900654-P). The microcontroller then publishes a feedback signal to a ROS topic to indicate that the valve was triggered. The central and vision modules then finally read this message to update the spray count and clear any spray schedules that were within the previous spraying time, respectively.

Three key innovations differentiate this sprayer prototype from other existing designs:The sprayer utilized a modular hardware and software architecture, making the design scalable and reconfigurable. The manually pushed prototype in the test was limited to three modules with the consideration of human power. The same design can be easily expanded to unlimited modules as long as the power and maneuverability are allowed by the prime mover, such as a tractor or unmanned vehicle.The vision modules used the virtual crop and weed detection bounding box to estimate the travel velocity in a local coordinate system. In this “what you see is what you detect” approach, the vision module can combine plant detection and velocity estimation. It can also easily correct any error with real-time feedback from the incoming video streams. Compared with wheel encoders [10] or global positioning systems with real-time kinematics [25], the vision module could be potentially more accurate, faster to obtain feedback, and more capable to accommodate uneven terrain.The effective spray regions covered by the nozzles were positioned away from the velocity estimation region, as illustrated in Figure 3. This approach will decrease the need for computing power while increasing the permissible time delay between detection and spraying to allow for a higher sprayer moving speed than when the detection, velocity estimation, and sprayer regions coincide.

### 2.2. Experimental Field

The image dataset collection and field testing were performed in the agricultural field in South Jersey Technological Park of Rowan University, Glassboro, NJ, USA, as shown in Figure 4. The farm was located at 39°43′08.1″ N and 75°08′52.5″ W and was planted with Pioneer-brand soybean variety during the first week of June 2022. The broadcast sowing of soybean on the experimental fields was performed at approximately 395,000 seeds per hectare.

### 2.3. SSD-MobileNetV1 Training and Validation

Several videos were recorded from 30 June to 8 July 2022, at approximately 4-day intervals, using two video capture devices (Apple iPhone 11 and Logitech StreamCam). The videos were recorded at different camera heights, angles, times of day, and growth periods. Overall, 877 RGB images were extracted from the captured videos and were then annotated using LabelImg [35]. A binary classification, “soybean” and “weed”, was implemented during annotation, and all non-soybean plants were grouped into the “weed” category. Figure 5 illustrates sample non-soybean plants found in the test field. The final dataset contained 5080 and 6934 instances of soybeans and weeds, respectively, and was randomly divided into 80% training and 20% validation.

The SSD-MobileNetV1 model was trained using an NVIDIA Jetson NX Xavier with Jetpack 4.6.1 and Pytorch SSD [36]. The training was performed for about 4000 epochs at an initial learning rate of 0.005, base learning rate of 0.0005, momentum of 0.9, weight decay of 0.00005, and batch size of 24.

The performance of the trained CNN model for soybean and weed detection was then evaluated using PASCAL VOC 2007 metrics. Precision (*P*), recall (*R*), average precision per class (AP), and mean average precision (mAP) at 0.5 intersection over union (IOU) threshold were used to describe the detection performance. Precision quantifies the percentage of the total detections that were correct relative to an object class, while recall indicates the percentage of the total class samples that were correctly detected. IOU represents the accuracy of the localization by comparing the generated bounding box against the annotated bounding box. $AP_{class}^{IOU}$ accounts for the classification and localization performance of the CNN model for a class by evaluating the area under the precision–recall curve at a specific IOU. Finally, $mAP^{IOU}$ is the average of AP for all object classes.

### 2.4. Field Testing

The field testing was performed on 11 July 2022 (28.9 °C, 11.1 km/h wind speed, 42–44% relative humidity) on three adjacent 0.5 m × 10 m rows of broadcast-seeded soybeans with randomly growing weeds. A video of the test plots was recorded and analyzed per frame to construct the 1 m resolution maps for the distribution and location of the weeds and samples in each test row. The weed population distribution for each row is shown in Figure 6a and were approximately 6.6, 17.8 and 11.8 weeds/m^2^ on the left, middle, and right test rows, respectively. In total, 30 target weeds ($N_{w}$) and 30 non-target soybeans ($N_{s}$) were randomly selected among the test rows, as shown in Figure 6b,c, respectively. Overall, 10 out of the 30 randomly selected soybeans had no adjacent weeds (Figure 6c), also referred to as soybean without weeds ($N_{s|wow}$). The performance testing was then performed in three trial runs.

#### 2.4.1. Targeting Performance

Target weeds and non-target soybeans were physically labeled to serve as visual references during the evaluation of the spray performance of the MAPS. The test videos were also recorded and carefully inspected per frame to accurately examine the targeting performance. Figure 7 shows a sample of the recorded frame.

The performance of the targeting system of the sprayer was then described in spraying precision ($P_{s}$) and recall ($R_{s}$). In weed spraying, correctly sprayed weeds were considered true positives (TP). Incorrectly sprayed soybeans were considered false positives (FP). Unsprayed soybeans were true negatives (TN), and unsprayed weeds were false negatives (FN). $P_{s}$ and $R_{s}$ were then calculated using Equations (Equation 1) and (Equation 2), respectively. $P_{s}$ describes the reliability of the sprayer to target weeds instead of soybeans. $R_{s}$ represents the accuracy of the sprayer in targeting all weeds. On the other hand, the wrong spraying rate (WS) was calculated using Equation (Equation 3) and depicts the incorrect spraying rate of non-target soybeans. Lastly, due to the proximity of soybean samples to weeds, the non-targeting rate (NT), or the fraction of unsprayed soybeans without adjacent weeds, was also determined using Equation (Equation 4).
(1)$$P_{s}=\frac{TP}{TP+FP}$$
(2)$$R_{s}=\frac{TP}{TP+FN}$$
(3)$$WS=\frac{FP}{TN+FP}$$
(4)$$NT=\frac{TN}{N_{s|wow}}$$

#### 2.4.2. Spray Volume Reduction

The average variable spray volume in each row ($Q_{row|variable}$) was estimated by calculating the product of the average number of actuation of the solenoid valve ($N_{valve|row}$), nozzle flowrate ($Q_{nozzle}$), in L/min, and nozzle opening time ($t_{spraying}$), in seconds, as shown in Equation (Equation 5).
(5)$$Q_{row|variable}=N_{valve|row}\times Q_{nozzle}\times t_{spraying}$$

The sprayer was pre-calibrated and tested the flowrate of each nozzle using ASAE EP367.2 MAR1991 (R2017) standard [37], yielding $Q_{nozzle}$ of 1.6 $\frac{L}{\min}$. The average uniform spray volume for each row ($Q_{row|uniform}$) was then the product of the $Q_{nozzle}$ and spraying trial time ($t_{trial}$), as shown in Equation (Equation 6). Finally, based on the pre-calibrated flow rate of the nozzle, the spray volume reduction (SVR) was calculated using Equation (Equation 7).
(6)$$Q_{row|uniform}=Q_{nozzle}\times t_{trial}$$
(7)$$SVR=\frac{Q_{row|uniform}-Q_{row|variable}}{Q_{row|uniform}}$$

## 3. Results and Discussion

### 3.1. CNN Model Performance

Figure 8 illustrates the precision–recall curve of the trained CNN model at 0.5 IOU threshold and shows that the model detected soybean better than weeds. At 68% recall and 0.5 IOU threshold, the trained model had an equivalent 71% and 90% precision in detecting weeds and soybeans, respectively. Compared to the precisions obtained in detecting weed patches in aerial images of row-seeded soybean using SSD (65%) and Faster R-CNN (66%) at 0.5 IOU and 68% recall [21], our model exhibited higher weed detection precision (71%). This difference was most likely due to differences in the experimental setups, such as the early-season time of entry when there was less overlap among soybeans and weeds, and a closer field of view using ground images in our study. In weed spraying, a high detection recall lowers the rate of unsprayed weeds, while high precision increases chemical savings by preventing unwanted nozzle openings due to false detection. Since the high rate of unsprayed weeds can potentially reduce crop yield, we prioritize detection recall over precision, as the former directly affects the effectiveness of weed control.

Table 2 summarizes the detection performance of the SSD-MobileNetV1. Validating the CNN model yielded 76.0% $mAP^{0.5}$. Running the model on the vision module with the custom algorithms resulted in a 19-fps effective inference speed. Compared to a sprayer with YOLOv3-tiny running on an NVIDIA GTX 1050 mini PC [24], our design had similar detection accuracy but 40% slower than their configuration (76.4% $mAP^{0.5}$ and 31.5 fps). However, a mini pc with NVIDIA 1050 was also about twice the cost of three NVIDIA Jetson Nano 4 GB and one Raspberry Pi 4B 4 GB.

Figure 9 shows a sample soybean and weed detection of the trained model. The image shows most small soybeans and weeds were undetected at a 0.5 confidence threshold. This behavior of SSD-MobileNet-based models was similarly observed in a previous study by the authors where the SSD-MobilenetV2 model had low detection confidence for small objects [38]. However, this situation may not be consequential in actual spraying scenarios, as small weeds would have difficulty competing with large soybeans for resources. Crops could tolerate a level of weed presence without quality and yield reductions [39]. According to the study of Datta et al. [40], in managing weeds using crop competition in soybean, weeds that emerged after the soybean emergence stage tend to have a lower competitive index than weeds that emerged before or during soybean emergence, and the closing of the canopy can suppress weeds that escaped herbicide application.

### 3.2. Targeting Performance

Table 3 summarizes the field spraying performance of the MAPS and shows the successful spraying of all sampled target weeds. This high targeting performance of the sprayer can be attributed to the following: (1) operating within the maximum velocity and (2) queuing multiple spray schedules. Based on our previous study on the effect of travel velocity, inference speed, and camera configuration on missed detections [38], the test velocity was within the maximum operating velocity of the vision module (18.05 m/s), preventing undetected weeds due to gaps between processed frames. Further, despite only having 76% $mAP^{0.5}$, traversing at 0.69 m/s and 19 fps allowed the vision module to detect plants on multiple frames. The weed targeting algorithm only required a single frame with correct weed detection and velocity estimate to calculate an accurate spray schedule. If a weed was incorrectly detected in a frame, the weed could still be possibly detected in succeeding frames.

Comparing the results with other studies, our configuration achieved higher spraying recall (100%) than a similar CNN-based precision sprayer for soybeans with Mask R-CNN (78%) and YOLOv3 (20%) that utilized a desktop computer with NVIDIA GeForce GTX 1050 [23]. Queuing multiple spray schedules and positioning the nozzle away from the frame center might have caused the difference in our results, as [23] had higher detection precision (≥88%) than our configuration. Similarly, the high spraying recall, despite low detection performance, was also observed by Ruigrok et al. [27]. Despite having 57% precision and 84% recall in detecting weeds, their system still achieved 96% average spraying recall by requiring a single correct detection among the processed frames. However, the 4 fps effective inference speed of their system was approximately four times slower than our configuration. This situation resulted in more frames available in our system for inferencing, and most likely contributed to the higher spraying recall than that of Ruigrok et al. [27].

On the other hand, our average spraying precision (57.32%) was relatively low compared to the results of Partel et al. [25] (78%) and Liu et al. [26] (96.67%). This difference was most likely caused by the broadcast-seeded layout and higher weed population in our tests compared to the mentioned studies, which had a row-seeded layout with approximately 1 m hill spacing and up to 30 weed population. Evaluating the results of our tests, the broad coverage of the nozzle, unintended sprays, and the natural proximity of targets and non-targets caused the low spraying precision. As described in Figure 6c, only 10 sample soybeans had no adjacent weeds. This high rate of FP of the sprayer caused by coarse nozzle resolution and proximity of non-targets to targets was also observed Ruigrok et al. [27]. Their system sprayed 50% of non-targets, on average, due to proximity to targets. On the other hand, the MAPS sprayed approximately 74% of non-targets.

For each spray, the covered land is a shape of a stretched eclipse with a width of the $2h\times tan\theta_{T}$ and a length of $t_{spraying}\times v_{vehicle}+2h\times tan\theta_{L}$, where *h* is the height of the nozzle from the ground, $\theta_{T}$ is the transverse spraying angle of the nozzle, $t_{spraying}$ is the spraying time, $v_{vehicle}$ is the velocity of the MAPS, and $\theta_{L}$ is the longitudinal spraying angle of the nozzle. Because the spraying nozzle is located at the center of the vehicle, the influence of the wind can be ignored since the side of the frame can be covered. To improve the precision, we could choose a nozzle with a narrow spray pattern or reduce the valve opening time for each spray in the future.

Figure 10 then illustrates sample detection scenarios showing the labeled target and non-targets during the experiment. An ideal scenario is shown in Figure 10a, where the weed at the center of the frame only had another weed in proximity along the horizontal axis of the frame. Furthermore, the soybeans at the bottom of the frame also had no adjacent weeds, resulting in them not being sprayed. This case was also observed in Figure 10c, where the soybeans at the top of the frame were also not sprayed. Figure 10b, on the other hand, shows a non-target soybean in the middle of the frame with weeds growing on its left side. This non-target soybean was unintentionally sprayed. Lastly, Figure 10d shows a non-target soybean incorrectly detected as a weed causing it to be sprayed.

The sprayer successfully avoided spraying 76.67% of soybeans without adjacent weeds, on average. Still, due to inaccuracy and extreme fluctuations in weed location and travel velocity estimates, the multiple spread-out spraying schedules for a weed and incorrect detections also caused unintentional spraying of 10% to 30% of soybeans without adjacent weeds. If soybean is incorrectly detected as a weed, this situation results in incorrect targeting. Moreover, the algorithm measures the distance traveled by a detected plant between two consecutive frames. If a wrong detection occurs in the succeeding frame, the error contributes to inaccurate velocity estimates and may cause the valve to open when soybeans are directly below the nozzle.

Overall, the targeting performance of the MAPS maintained similar performance while operating at up to three times the velocity of existing CNN-based sprayers. The average velocity of MAPS was 0.69 m/s (2.5 km/h). Due to the rough terrain, the recorded velocity varied between 0.53 m/s (1.9 km/h) and 0.83 m/s (3.0 km/h). The theoretical velocity can reach 3.54 m/s (12.7 km/h), which was only limited by the current push-style configuration. Among the compared studies, Liu et al. [26] reported 0.28 m/s (1.0 km/h) during the field test on real plants. Sabóia et al. [23] reached 0.5 m/s (1.8 km/h) at low targeting performance. On the other hand, Farooque et al. [24] reported 1.38 m/s (5.0 km/h) on the field test of their developed sprayer, but the field targeting performance was not quantified.

### 3.3. Spray Volume Reduction

Table 4 summarizes the variable sprayed volume of the MAPS on each row with different weed populations. The results showed that the average count of the solenoid valve opening increases with the weed population, an indicator of the variable spraying capability. The left row had the lowest weed population of 33 and average spray instances of 38. Similarly, the middle row with the highest weed population of 89 had the highest average spray count of 57. The disproportional increase is due to the high density of the weeds, where one single spray could cover two or more weeds with the movement of the MAPS. Theoretically, the 0.2 s spray duration and 0.69 m/s velocity would cover 0.138 m, which resulted in a maximum of 73 spray instances for a 10-m row fully covered with weeds. In contrast, the average space between weeds was only 0.112 m in the middle row. As a result, the $N_{valve|row}$ to the weed population per row ($N_{w|total}$) at the middle row was lower than that of the left and right.

The results also demonstrated that the sprayer had an average spray volume reduction of 38.96%, most likely representing the bare soil and soybean-only regions in the experimental field. The highest spray reduction (48.89%) was observed in the left row, where the weed population was also the lowest. Consequently, the lowest volume reduction (23.56%) was observed in the middle row, which had the highest weed population. This result is consistent with our expectation, as the higher weed density needed more frequent sprays, which was closer to continuous spray. The lower the weed density, the higher the spray reduction will be. With the improved control of weeds after each spray, the reduction of total spray volume will also improve significantly. Nonetheless, direct comparisons with other studies were difficult, as the weed population of their test area was not reported. Both Farooque et al. [24] and Zanin et al. [16] reported similar spray volume reduction without the corresponding weed population.

## 4. Conclusions

The developed precision sprayer with CNN-based MVS and modular architecture successfully demonstrated its capability to target weeds and reduce spray volume in a broadcast-seeded soybean field. By using multiple edge devices to run the CNN model, the vision system achieved 19 fps and 76% $mAP^{0.5}$, resulting in the developed sprayer having similar targeting and spraying performance at a faster average velocity of 0.69 m/s (2.5 km/h) than CNN-based systems with known targeting performance. Furthermore, the field test also verified the variable spraying capability of the modular design, reducing spray volume by up to 48.89% in the experiments. Nonetheless, direct comparisons with existing CNN-based precision sprayers were difficult due to differences in the experimental setups and the unavailability of weed distribution. As demonstrated in our results, high weed density lowers spraying precision and spray volume reduction. When weed density is high, the likelihood that weeds are next to a crop is also very high, causing non-target crops to be unintentionally sprayed. Similarly, as the weed population increases, the number of spray instances of a precision sprayer also increases at a diminishing rate as it approaches the maximum spray instances equivalent to uniform spraying.

The broadcast-seeded layout and high weed density presented a challenging scenario for precision spraying. Despite these conditions, our CNN-based plant detection and vision-based velocity estimation proved to be doing well during operation regarding weed spraying recall and spray volume reductions. The spraying errors became secondary compared to the indirect spray caused by the wide effective spray region of the nozzle used in the test. Up to 90% of soybean samples without adjacent weeds were not sprayed during our trials. In contrast, all soybean samples with adjacent weeds were unintentionally sprayed. The former errors were caused by inaccurate plant detection and velocity estimation, while the latter could be attributed to the coarse resolution of the nozzle.

With the initial results of the field tests, improvements can be implemented in future design iterations to minimize unintentional spray on non-targets and improve the performance of plant detection and velocity estimation. First, we aim to optimize the number of nozzles per SU. While the increased number of nozzles can also increase agrochemical savings, it also increases overall system cost. Second, we plan to explore using the crop as the only reference for travel velocity estimation due to its higher detection performance and more regular distribution than weeds. Third, we plan to test the system with the aforementioned improvements at standard spraying velocity as a tractor implement or smart attachment for an agricultural robot with autonomous navigation. Finally, we shall explore cameras and other sensors for close-loop feedback of spray instances, as most precision sprayers generally use open-loop feedback.

## Figures and Tables

**Figure 1 sensors-22-09723-f001:**
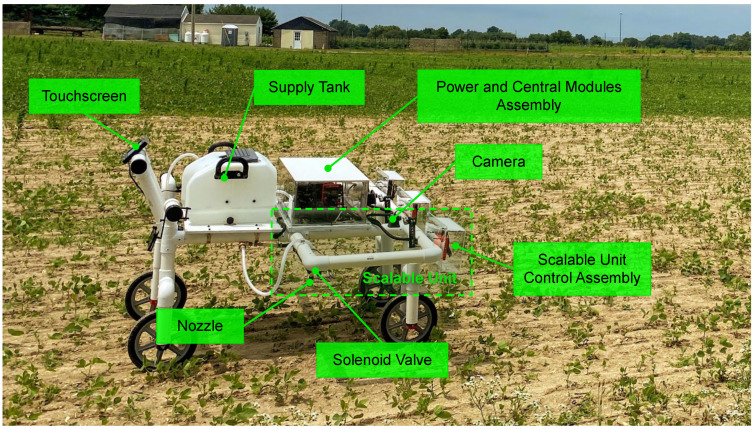
Components of the modular agrochemical precision sprayer mounted on a push-type frame. The precision sprayer had four key modules: central, power, vision, and sprayer. The modules were held together by a push-type reconfigurable frame.

**Figure 2 sensors-22-09723-f002:**
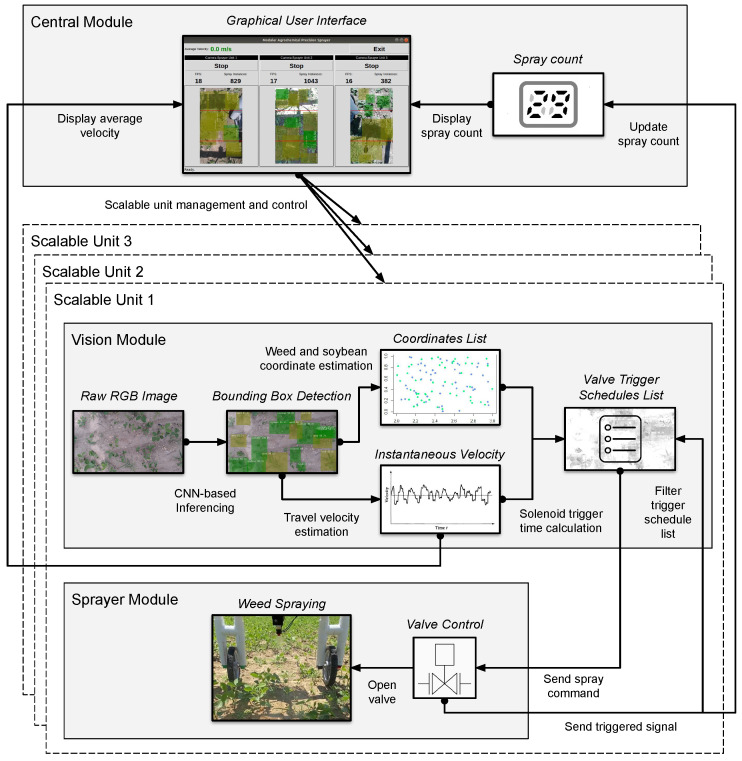
Work flow of a scalable unit of the precision sprayer.

**Figure 3 sensors-22-09723-f003:**
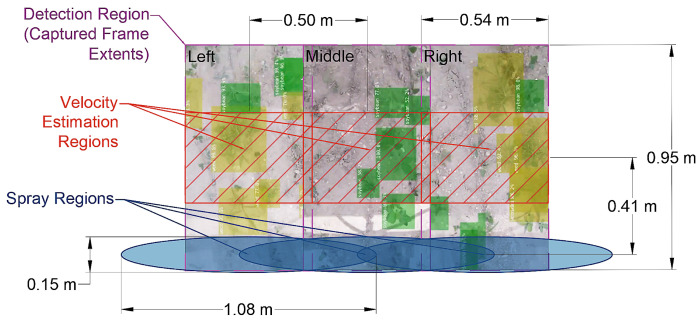
Layout of the positions of the detection, velocity estimation, and spray regions of the modular agrochemical precision sprayer.

**Figure 4 sensors-22-09723-f004:**
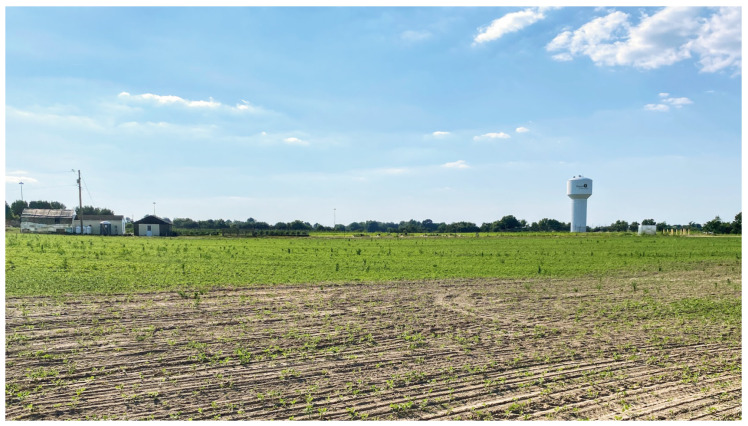
The experimental site for collecting soybean and weed images for CNN model development and precision spraying. The image was captured on 30 June 2022.

**Figure 5 sensors-22-09723-f005:**
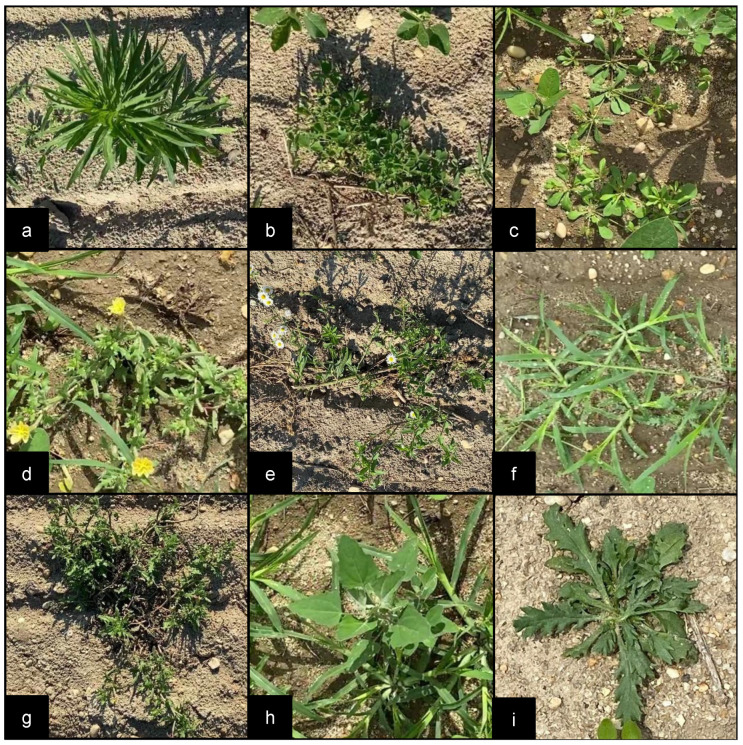
Sample images of target plants: (**a**) horseweed, (**b**) purslane, (**c**) carpet weed, (**d**) cut-leaved evening primrose, (**e**) hairy fleabane, (**f**) goosegrass, (**g**) ragweed, (**h**) lambsquarter, and (**i**) thistle.

**Figure 6 sensors-22-09723-f006:**
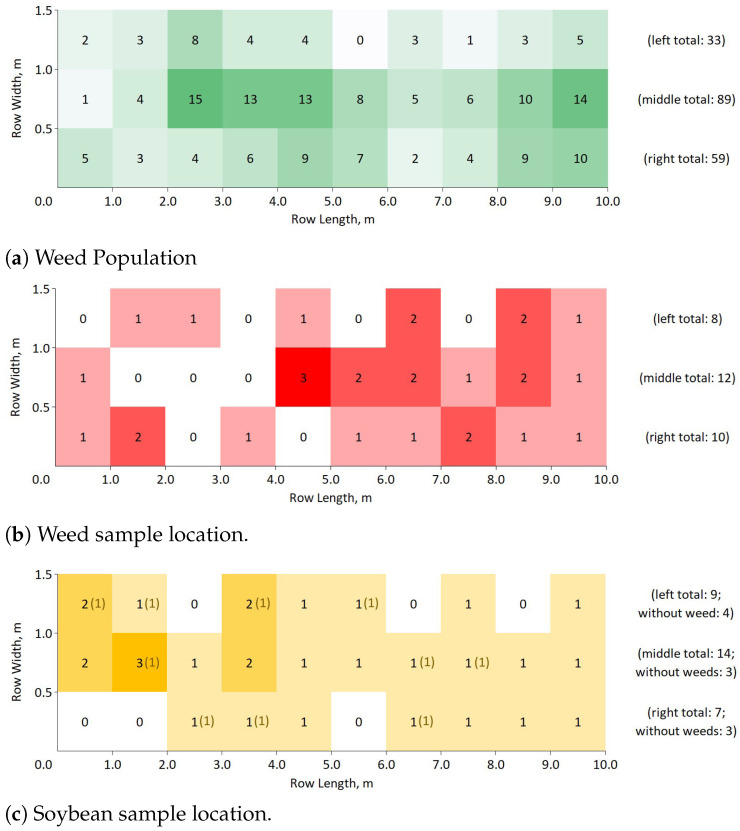
(**a**) Weed population, (**b**) weed sample location, and (**c**) soybean sample location in the three test rows. Values enclosed in parenthesis in soybean sample distribution represent soybean plants without adjacent weeds.

**Figure 7 sensors-22-09723-f007:**
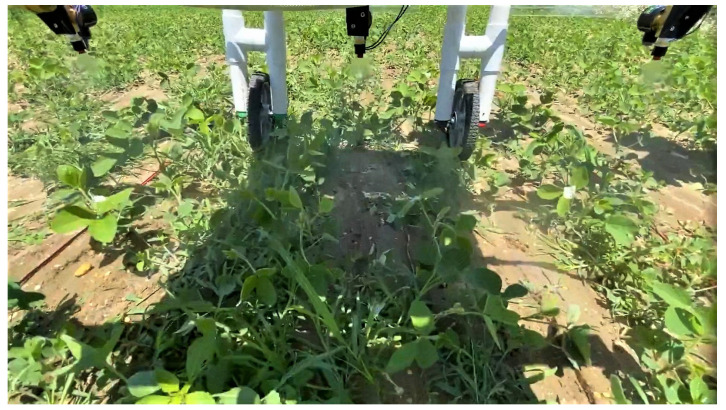
Sample recorded frame during the field test, showing labeled target and non-targets and spray instances.

**Figure 8 sensors-22-09723-f008:**
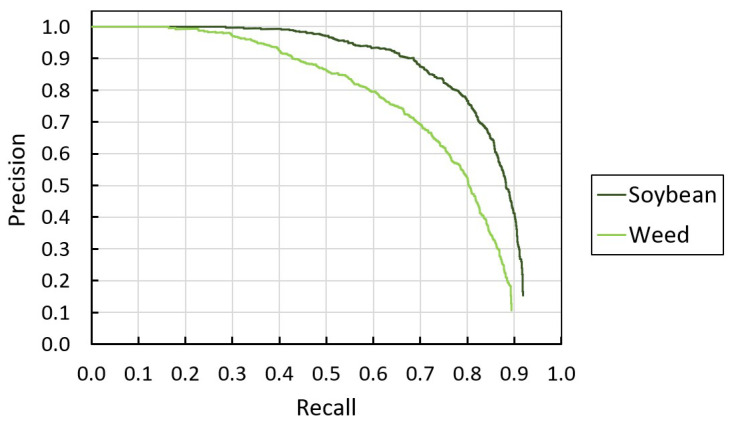
The precision–recall curve of the SSD-MobileNetV1 model on detecting soybean and weeds at 0.5 IOU threshold.

**Figure 9 sensors-22-09723-f009:**
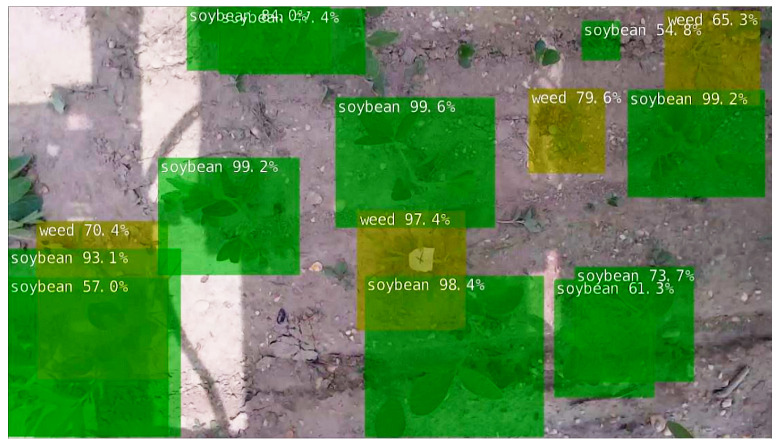
Sample soybean and weed detections of SSD-MobileNetV1 at 0.5 confidence threshold.

**Figure 10 sensors-22-09723-f010:**
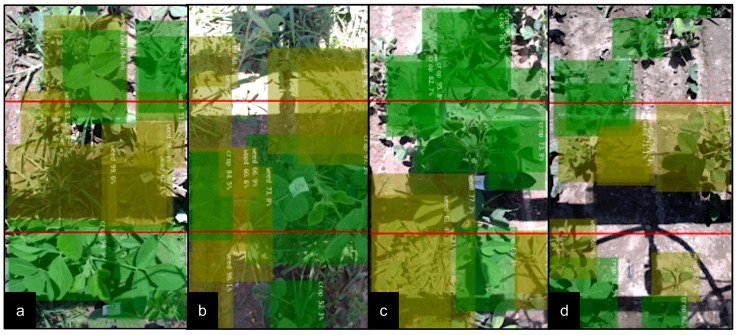
Sample detection scenarios during the experiment showing labeled targets weeds (yellow) and non-target soybeans (green): (**a**) labeled non-overlapping target and non-targets along the horizontal frame axis; (**b**) labeled non-target with an adjacent target; (**c**) labeled non-target with no adjacent target along the horizontal frame axis; and (**d**) non-target incorrectly detected as a target.

**Table 1 sensors-22-09723-t001:** The technical specification of the modular agrochemical precision sprayer with three scalable units.

Description	Value	Unit
Fluid Pressure	550	kPa
Nozzle Delivery Rate	1.6	L/min
Nozzle Spraying Time	0.2	s
Nozzle Spray Pattern Width	1.08	m
Nozzle Height	0.45	m
Nozzle Spacing	0.5	m
Effective Spray Width @ 50% Overlap	2.08	m
Max. Operating Ground Speed	3.54	m/s
Max. Theoretical Field Capacity	2.65	ha/h
Camera Resolution	1280 × 720	px
Average Inference Speed	19	fps
Power Consumption	160	W
Min. Operating Time	1.85	h

**Table 2 sensors-22-09723-t002:** Soybean and weed detection performance at 0.5 IOU threshold of SSD-MobileNetV1.

Class	${AP}^{0.5}$, %	${mAP}^{0.5}$, %	Inference Speed ${}^{a}$, fps
Soybean	81.4	76.0	19.0
Weed	70.6		

^a^ Running on the custom software of the precision sprayer using NVIDIA Jetson Nano 4 GB.

**Table 3 sensors-22-09723-t003:** Targeting performance of the MAPS at 0.69 ± 0.13 m/s.

Trial	TP	TN	FP	FN	$N_{s|wow}$	$P_{s}$, %	$R_{s}$, %	Ws, %	NT, %
1	30	7	23	0	10	56.66	100.00	76.67	70.00
2	30	7	23	0	10	56.66	100.00	76.67	70.00
3	30	9	21	0	10	58.82	100.00	70.00	90.00
Average	30	7.67	22.33	0	10	57.32	100.00	74.44	76.67

**Table 4 sensors-22-09723-t004:** Spray volume reduction of the modular CNN-based precision sprayer at 0.69 ± 0.13 m/s and 15 s average traverse time along soybean plots with different weed populations.

Row	$N_{w|total}$	$N_{valve|row}$	$N_{valve|row}/N_{w|total}$	$Q_{row|variable}$${}^{a}$, L	SVR${}^{b}$, %
Left	33	38.33 ±7.72	1.16	0.204 ±0.012	48.89
Middle	89	57.33 ±11.09	0.64	0.306 ±0.059	23.56
Right	59	41.67 ±10.21	0.71	0.222 ±0.051	44.44
All	181	137.33 ±22.54	0.76	0.732 ±0.120	38.96

^a^ Calculated at 0.2 s spray duration. ^b^ Calculated at 1.6 L/min continuous nozzle delivery rate.

## Data Availability

The data presented in this study are available on request from the corresponding author.

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
