# Peer review of "Evaluation of a CNN-Based Modular Precision Sprayer in Broadcast-Seeded Field"

_sensors, 2022, doi:10.3390/s22249723_

Round 1

Reviewer 1 Report

1. The accuracy recall curve of the convolutional neural network model after training is not entirely reasonable compared with SSD and Mask R-CNN, using the results of sowing and row sowing as evaluation metrics.

2. After the analysis of Figure 9, the authors stated that the analysis was unnecessary. I needed clarification on what the analysis focused on.

3. In the material and method, the evaluation index of the target performance requires to be clearly introduced.

4. In the comparison of accuracy of mAP0.5 threshold, there are two indicators for defining accuracy. , there are two indicators for defining accuracy. Which indicator is more critical in precision and recall rate should be apparently stated.

5. The experimental data analysis only includes the experimental data of the proposed method, and there is no comparative experiment. The experimental results of the literature directly cited are compared and analyzed. It may not be feasible.

Author Response

We sincerely thank you for your suggestions and comments on the improvement of our paper. We have included your suggestions and revised the statements to add clarity. The revised texts were highlighted in yellow in the manuscript for quick location. We have also attached a PDF to provide a point-by-point response to the comments and a summary of the revisions.

Reviewer 2 Report

This paper presents Evaluation of a CNN-Based Modular Precision Sprayer in Broadcast-Seeded Field.The work looks interesting, but some modifications of the manuscript are needed prior to publication. Some comments to improve the manuscript as follows; 1. In the Introduction, authors say that; "However, vision systems that analyze morphology and texture features  generally require a high image resolution, and detection algorithms are complicated and  computationally expensive for real-time systems" I do not agree with this content. If you say that, CNN is also computationally expensive, compared to many image processing based approaches. You can easily execute the many image processing techniques by using a Jetson nano hardware.   2. I believe that authors can use the models mentioned in the below papers for the same purpose since they are more light-weight models.    Small and Slim Deep Convolutional Neural Network for Mobile Device, REAF: Reducing Approximation of Channels by Reducing Feature Reuse Within Convolution.  Reviewer recommends discussing this in the introduction.   3. In your approach, plant detection and velocity estimation are not very accurate. There are many  methods in the literature that can be used for this detection and velocity estimation. Reviewer recommends applying approaches in the literature and making a comparison regarding this.     

Author Response

We greatly appreciate the suggestions to improve the technical soundness and add clarity to our statements in this manuscript. We attached a PDF of our responses to summarize the revisions made to the article. We increased the number of literature to improve the background of the study. The revised texts were in blue font in the journal article format to quickly locate the changes that we made.

Round 2

Reviewer 1 Report

I assumed that the revised manuscript has been properly modified according to my previous concerns and had met the eligibility standards of the journal.

Author Response

Thank you very much for all your input to improve our manuscript. We greatly appreciate all of them.

Reviewer 2 Report

N/A

Author Response

(The authors gave the same response as above.)
